# A critical assessment of the detailed *Aedes aegypti* simulation model Skeeter Buster 2 using field experiments of indoor insecticidal control in Iquitos, Peru

**Christian E. Gunning**[1,2], **Amy C. Morrison**[3,4], **Kenichi W. Okamoto**[5], **Thomas W. Scott**[6], **Helvio Astete**[3], **Gissella M. Vásquez**[3], **Fred Gould**[2,7], **Alun L. Lloyd**[7,8]*

**1** Odum School of Ecology, University of Georgia, Athens, Georgia, United States of America, **2** Department of Entomology and Plant Pathology, North Carolina State University, Raleigh, North Carolina, United States of America, **3** Department of Virology and Emerging Infections and Department of Entomology, U.S. Naval Medical Research Unit No. 6, Lima and Iquitos, Peru, **4** Department of Pathology, Microbiology, and Immunology, School of Veterinary Medicine, University of California, Davis, California, United States of America, **5** Department of Biology, University of St. Thomas, St. Paul, Minnesota, United States of America, **6** Department of Entomology and Nematology, University of California, Davis, California, United States of America, **7** Genetic Engineering and Society Center, North Carolina State University, Raleigh, North Carolina, United States of America, **8** Biomathematics Graduate Program and Department of Mathematics, North Carolina State University, Raleigh, North Carolina, United States of America

* alun_lloyd@ncsu.edu

**Data Availability Statement:** The authors confirm that all data underlying the findings are fully

## Abstract

The importance of mosquitoes in human pathogen transmission has motivated major research efforts into mosquito biology in pursuit of more effective vector control measures. *Aedes aegypti* is a particular concern in tropical urban areas, where it is the primary vector of numerous flaviviruses, including the yellow fever, Zika, and dengue viruses. With an anthropophilic habit, *Ae. aegypti* prefers houses, human blood meals, and ovipositioning in water-filled containers. We hypothesized that this relatively simple ecological niche should allow us to predict the impacts of insecticidal control measures on mosquito populations. To do this, we use Skeeter Buster 2 (SB2), a stochastic, spatially explicit, mechanistic model of *Ae. aegypti* population biology. SB2 builds on Skeeter Buster, which reproduced equilibrium dynamics of *Ae. aegypti* in Iquitos, Peru. Our goal was to validate SB2 by predicting the response of mosquito populations to perturbations by indoor insecticidal spraying and wide-spread destructive insect surveys.

To evaluate SB2, we conducted two field experiments in Iquitos, Peru: a smaller pilot study in 2013 (S-2013) followed by a larger experiment in 2014 (L-2014). Here, we compare model predictions with (previously reported) empirical results from these experiments. In both simulated and empirical populations, repeated spraying yielded substantial yet temporary reductions in adult densities. The proportional effects of spraying were broadly comparable between simulated and empirical results, but we found noteworthy differences. In particular, SB2 consistently over-estimated the proportion of nulliparous females and the proportion of containers holding immature mosquitoes. We also observed less temporal variation in simulated surveys of adult abundance relative to corresponding empirical

available without restriction. Code and documentation for SB and SB2 is available at https://github.com/helmingstay/SkeeterBuster. Data are available at https://osf.io/jsfn8/.

**Funding:** This research was supported by funding from the National Institute of Allergy and Infectious Diseases under award numbers R01-AI091980 (ACM, FG, TWS, and ALL) and R01-AI139085 (ACM, FG, and ALL), and from the National Science Foundation under award number RTG/DMS-1246991 (ALL). The funders had no role in study design, data collection and analysis, decision to publish, or preparation of the manuscript.

**Competing interests:** The authors have declared that no competing interests exist.

observations. Our results indicate the presence of ecological heterogeneities or sampling processes not effectively represented by SB2. Although additional empirical research could further improve the accuracy and precision of SB2, our results underscore the importance of non-linear dynamics in the response of *Ae. aegypti* populations to perturbations, and suggest general limits to the fine-grained predictability of its population dynamics over space and time.

## Author summary

*Aedes aegypti* is commonly found in tropical urban areas, and is the primary vector of several serious human pathogens, including yellow fever, Zika, and dengue. As such, control of *Ae. aegypti* presents a pressing global public health concern, particularly in low-resource settings. Previous work has used biologically-detailed simulation models (e.g., Skeeter Buster) to predict equilibrium *Ae. aegypti* population dynamics over space and time. Here we present an improved model, Skeeter Buster 2 (SB2), that includes location-specific sampling and spraying events. We use SB2 to simulate, as closely as possible, two field trials of non-residual insecticidal spraying in Iquitos, Peru during 2013 and 2014. Finally, we critically assess SB2's ability to predict non-equilibrium responses of *Ae. aegypti* populations to vector control efforts by comparing simulations and empirical observations.

Overall, we found that the effects of spraying were broadly comparable between simulated and empirical results, including rapid post-control recovery. Notably, we observed less temporal variation in simulated adult abundance than in empirical observations. Our results indicate the presence of ecological heterogeneities and/or sampling processes not captured by SB2, and suggest limits to the fine-grained predictability of *Ae. aegypti* population dynamics over space and time.

## Introduction

Despite widespread prevention efforts, the public health impact of dengue has increased significantly over the past 50 years, both in overall burden and expanded geographic distribution. The most common interventions for disease prevention to date have focused on dengue's primary vector, *Aedes aegypti* [1–4]. And, while ongoing efforts aimed at the development of vaccines and novel vector control strategies show potential, broadly effective tools are not expected to be available for public health application in the short-term [5]. Unfortunately, vector control as currently practiced has yielded inconsistent and generally disappointing results [2, 6–8] in part due to gaps in our understanding of *Ae. aegypti's* ecology and life history [9–12].

*Ae. aegypti* is adapted to an anthropophilic lifestyle: adults live in close association with humans, females feed almost exclusively on human blood [13, 14], and larvae develop in water-filled containers, mostly located in and around households [9, 15, 16]. As such, transmission of dengue virus is largely driven by interactions between *Ae. aegypti* and humans within and around human dwellings. To better understand these processes, numerous biological process models have been developed to simulate *Ae. aegypti* population dynamics [17–20] and transmission of dengue virus [21]. Yet we lack critical assessments of existing models' ability to predict non-equilibrium *Ae. aegypti* population dynamics [22, 23] or dengue

epidemiology [24, 25]. Factors complicating predictions of *Ae. aegypti* population dynamics include heterogeneity of vector biotic and abiotic habitat, as well as parameterization of relevant biological processes, such as mosquito life history. Although recent simulations of *Ae. aegypti* have included insecticidal interventions [21, 26], to date, models of *Ae. aegypti* dynamics have not been tested to determine whether they can reliably predict the impacts of vector control measures.

In previous work we developed Skeeter Buster (SB): a biologically detailed, agent-based stochastic simulation model of *Ae. aegypti* population dynamics that aimed to address the dual questions of spatiotemporal detail and empirical realism [18]. SB was built upon a biologically detailed model of *Ae. aegypti* (CiMSIM) that parameterized biological processes that researchers and practitioners considered critical to predicting *Ae. aegypti* population dynamics [17, 22, 27]. Because CiMSIM did not include any spatial structuring or stochasticity, it did not address the expected heterogeneity of *Ae. aegypti* populations in urban areas, the impacts of low densities of adult mosquitoes, or the consequences of mosquito movement across a landscape of separated, discrete habitats (i.e., human dwellings). SB was parameterized with values for Iquitos, Peru, and its performance was initially evaluated for correspondence with the observed, unperturbed dynamics of *Ae. aegypti* in Iquitos [23, 28].

Vector suppression efforts necessarily perturb mosquito populations, whose subsequent responses must be considered in the design and implementation of control strategies [29–31]. To better understand the response of *Ae. aegypti* populations to perturbations, we designed a series of two field experiments in Iquitos using insights gained from SB. A smaller pilot experiment was conducted in 2013 (henceforth S-2013), followed by a larger and more comprehensive experiment conducted in 2014 (henceforth L-2014) [32]. During each experiment, we used indoor ultra-low volume (ULV) pyrethroid insecticide spraying of individual households to suppress *Ae. aegypti* populations in a region of the study area, and also included an unsprayed buffer region. These experiments employed non-residual ULV spraying and did not incorporate larvicide in order to differentiate the immediate impacts of suppression efforts from subsequent population dynamics. The resulting dataset represents one of the most detailed accounts available of the spatiotemporal impact of area-wide spray interventions at a neighborhood spatial scale.

Spatial heterogeneity of suppression efforts and responses by mosquito populations can also impact control strategies [33]. To better capture spatial heterogeneities of these two field experiments, we revised the spatial structure of SB. The resulting model, Skeeter Buster 2 (SB2), takes as input the specific dates and locations of ULV spraying and sampling (i.e., household surveys of larval habitat and adult abundance), allowing us to directly simulate each field experiment and then compare simulated observations to experimental survey results. We note that *Ae. aegypti* population dynamics remain unchanged in SB2, and that SB2 is capable of reproducing the grid-based spatial configuration used by SB.

Here we assess the performance of SB2 against empirical observations of the S-2013 and L-2014 field experiments (which were themselves explicitly designed for this purpose). Our primary goal is model validation in the face of non-equilibrium dynamics; to avoid circularity, we do not explicitly *fit* our model here, instead relying on SB's previous parameterization in Iquitos [18, 23]. We first present model predictions of the expected underlying dynamics of mosquito populations in order to assess their response to ULV spraying and destructive sampling (i.e., removal of sampled adults and immatures). We next compare simulated observations with empirical observations. We identify areas of concordance between empirical and simulated results, and explore predictions of the model that diverge from the empirical data. Ultimately, we seek to predict when, and for how long, indoor space spraying reduced mosquito populations, and to identify key processes that A) limit the efficacy of indoor ULV insecticidal

vector control, and B) drive the post-spraying recovery of mosquito populations. We also discuss noteworthy sources of uncertainty, and address the usefulness of SB2 (and similar models) in future vector control research.

## Methods

### Iquitos, Peru

Iquitos is the largest urban center in the Department of Loreto, situated in the Amazon Basin of northeastern Peru. Citywide vector monitoring and control efforts have been ongoing in Iquitos since 2002 [15, 34–37]. Here we focus on two field experiments that were conducted in 2013 and 2014 in separate neighborhoods in the Iquitos district in the city of Iquitos (Fig 1): an initial smaller pilot study (S-2013) and a larger follow-up (L-2014). These experiments evaluated the impact of non-residual, indoor ultra-low volume (ULV) pyrethroid spraying (henceforth spray or spraying) on *Ae. aegypti* populations. Each experiment was spatially configured to provide a central region that was sprayed (the **spray sector**) surrounded by an unsprayed outer region (the **buffer sector**). Here we use SB2 to simulate, as closely as possible, the

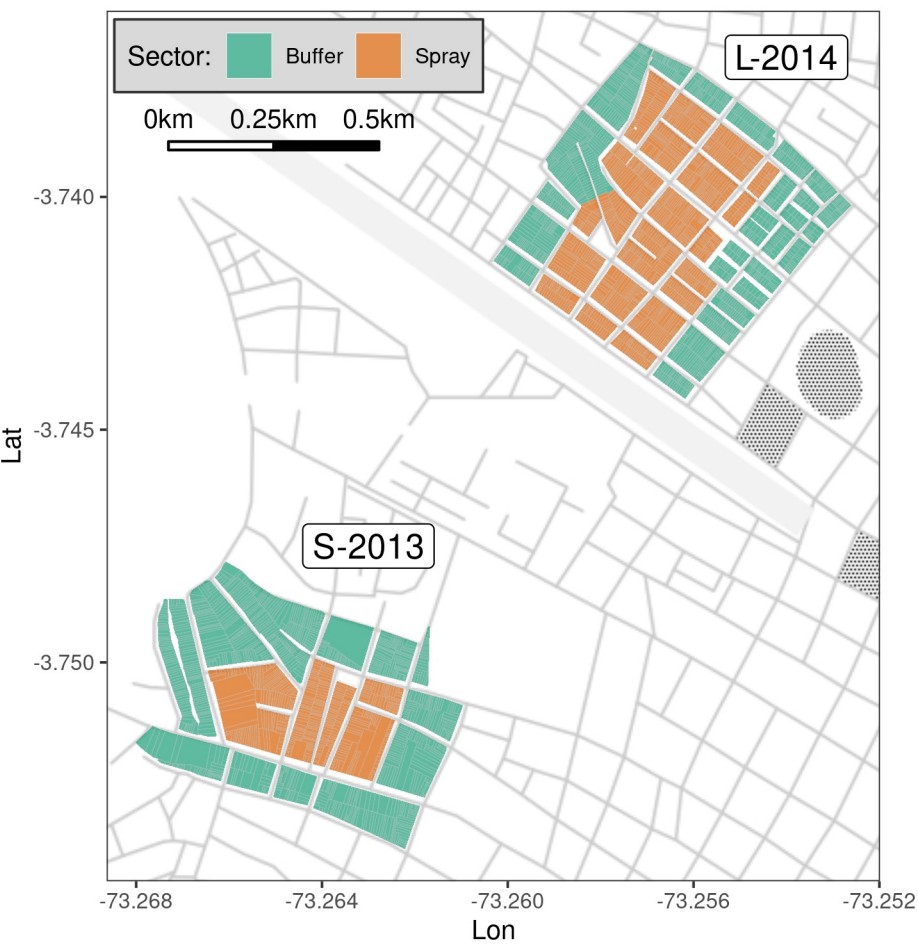

**Fig 1. Map showing study areas within Iquitos, Peru.** A smaller pilot study was conducted during spring and summer of 2013 (S-2013), followed by a larger follow-up study from January through October of 2014 (L-2014). Color shows sector. The L-2014 study area bordered an abandoned air strip on its northwest edge. Base map by Stamen Design, under CC BY 3.0; data by OpenStreetMap, under ODbL (http://maps.stamen.com/#toner).

empirical details of each experiment, including regional weather, adult and immature surveys, and spray interventions. These simulations allow us to critically assess SB2 by directly comparing simulated and empirically observed adult mosquito abundances over time and space.

Details of the study area, experimental design, and observed outcomes, along with a detailed ethics statement, are provided in Gunning et al. [32]; a brief overview is provided below.

### Field surveys

Field workers attempted to survey every study house exactly once in each experimental time period by conducting a methodical, full circuit of the study area. These survey **circuits** typically lasted 2–4 weeks (up to six weeks in L-2014, see S1 Table in Gunning et al. [32]). During adult surveys, trained surveyors searched for and collected all adult mosquitoes from in and around sampled houses using powered Prokopack aspirators [38]). We enumerated the number of adult mosquitoes per house by species. In addition, the parity of adult *Ae. aegypti* females was assessed post-collection by dissection and inspection of ovaries [39]. Container surveys recorded all identifiable water-holding containers (i.e., potential larval habitat) in and around each surveyed house (S1 Fig), including container type, dimensions, location (inside/outside), and water fill patterns (passive rain-filled, active rain-filled via roof or rain gutter, manually filled by resident). The presence and number of *Ae. aegypti* eggs, larvae, and pupae in each container were recorded, and positive containers were then emptied of water and immature insects. Note that field workers destroyed and/or removed all sampled insects (adults and immatures) from the surveyed house. For additional details regarding field surveys, please see Getis et al. [34] (container surveys) and Morrison et al. [40] (adult surveys).

Each experiment commenced with an initial baseline circuit (C1, 65–72% of houses successfully surveyed) that we used as a pre-intervention reference. Most circuits included adult and container surveys, but only adult surveys were conducted during spray circuits (see also Figures 2 and 3 in Gunning et al. [32]).

### Spray intervention

During each field experiment, scheduled ULV spraying of the *spray sector* was conducted in a set of six **spray cycles**, with each cycle lasting approximately one week. During each spray cycle, teams visited and attempted to spray every house in the spray sector exactly once. Adult surveys of sprayed houses were carried out either immediately prior to spraying (S-2013) or 1–4 days after spraying (L-2014, see also Figure 2 in Gunning et al. [32]). Taken together, the six scheduled spray cycles and corresponding surveys constitute a single circuit per year: S-2013 C2 and L-2014 C6. During L-2014, an additional unscheduled, emergency spray cycle (C3) was conducted citywide (i.e., in both sectors) by the Ministry of Health (MoH) in response to an impending dengue epidemic.

Pyrethroid insecticides included alphacypermetrin (S-2013) and two formulations of cypermethrin (L-2014). Sprays were applied using Solo or Stihl backpack sprayers with settings adjusted for ULV application, or Colt hand-held ULV sprayers. During each spray cycle, teflon treated slides were placed in 2 randomly selected houses and retrieved 1-hour post-spray; droplet size was then measured, and droplets were counted in a 1 $cm^2$ square.

During scheduled experimental spraying, small screened cages, each containing 25 adult female *Ae. aegypti* from a recently collected laboratory colony (Gunning 2018), were placed in randomly-selected houses immediately prior to spraying as a bioassay to estimate spray efficacy based on percent mortality of susceptible mosquitoes in the cages. When mortality was <80%, equipment was recalibrated to ensure proper spray function on subsequent days.

### Simulation overview

Both SB and SB2 simulate *Ae. aegypti* population dynamics according to the best available evidence regarding *Ae. aegypti* life history and population biology. We note that, with a few key exceptions (discussed below), SB2 exactly reproduces the behavior of SB (see also S1 Text). Detailed discussion of SB design, implementation, and parameterization are provided in Magori et al. [18] (e.g., Figs 1–5), with additional context provided by Focks et al. [17], Xu et al. [28], and Legros et al. [23].

We offer here a brief overview of model structure. Both models incorporate a spatially explicit configuration of houses. Within houses, water-filled containers that offer larval habitat are also simulated, including container food input, water level, and water temperature. These models simulate individual adult female *Ae. aegypti*, and cohorts of adult males and immature stages: eggs, larvae, and pupae. Adult *Ae. aegypti* mosquitoes (henceforth adults) dwell within and move among houses, which contain larval habitat (containers). Adult movement depends on the availability of mates and larval habitat (we assume human hosts are available in all houses). Adult females mate and subsequently lay eggs in water-filled containers, where container oviposition choice depends on container volume and the presence of a cover or lid, and larval survival development depends on food availability within containers. Survival of all life stages depends on temperature. Survival of adults and eggs also depends on water vapor pressure deficit (VPD, a function of temperature and relative humidity), with mortality increasing at low VPD.

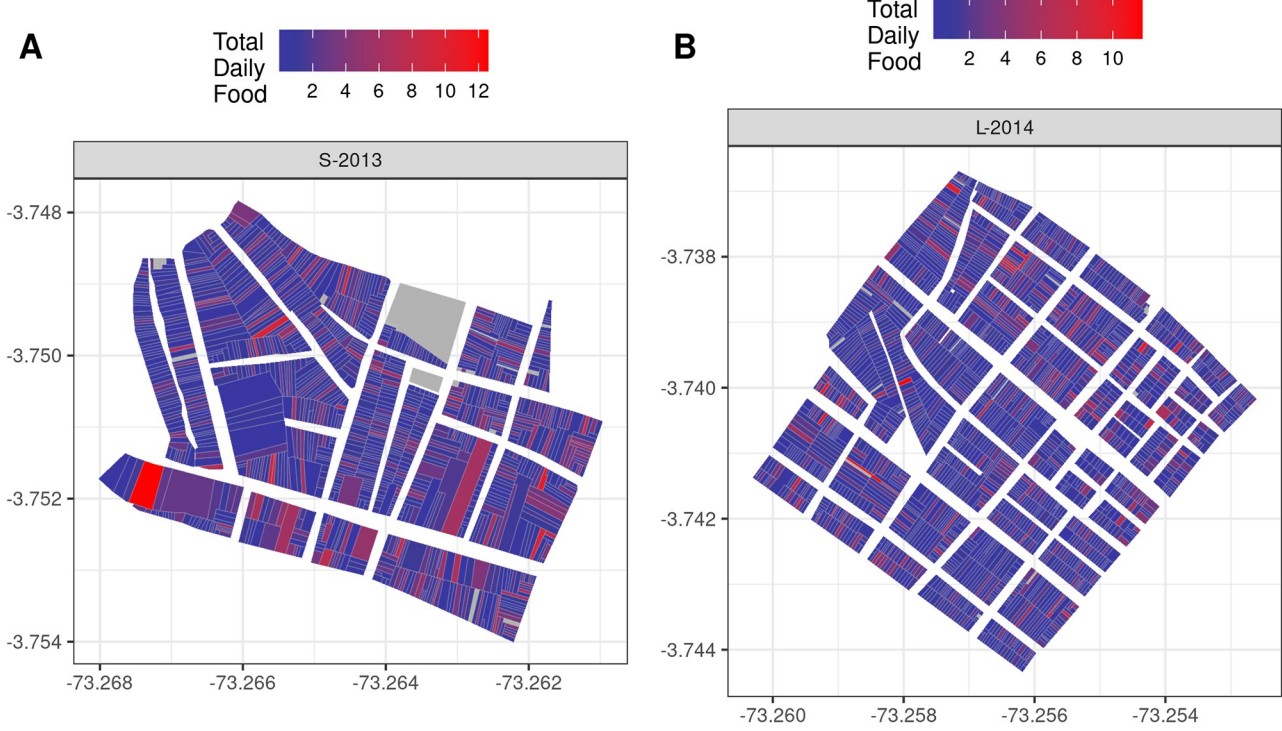

**Fig 2. Details of simulation spatial configuration.** Each polygon contains 1 or (rarely) more houses. Houses are grouped by blocks, which are separated by streets. Adult local migration occurs between neighboring houses within each block (see text for details). Color shows total daily food input per polygon. Polygons with no surveyed containers are shown in gray. See also S1 Fig.

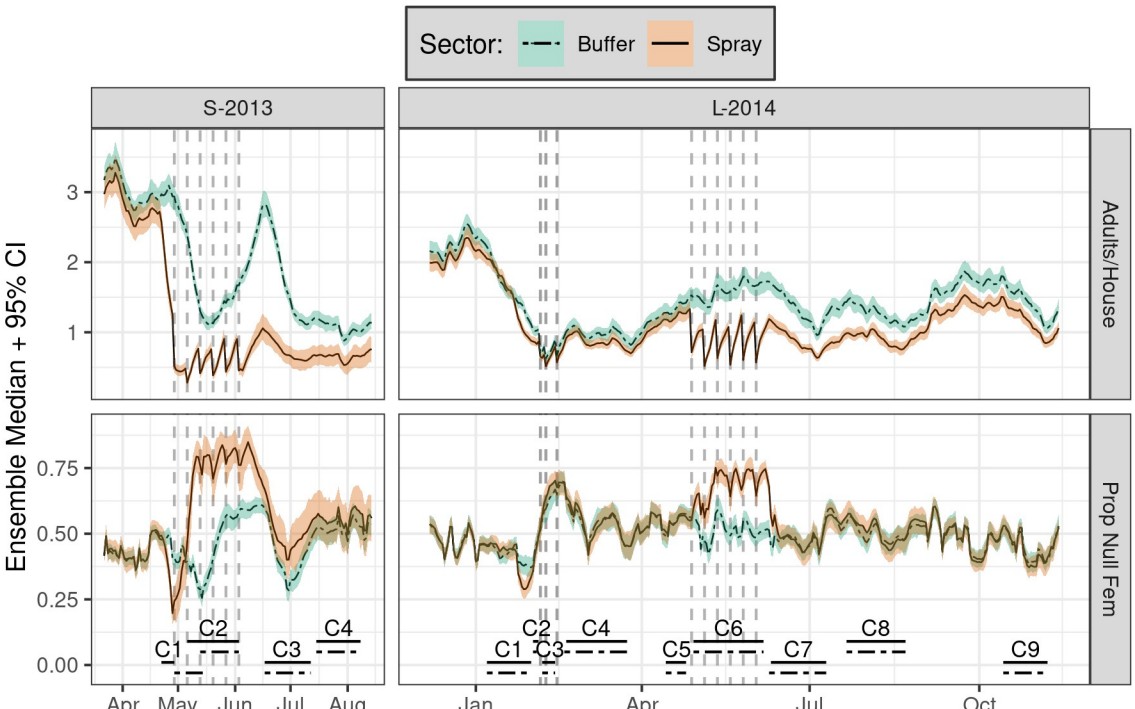

**Fig 3. Daily time series of simulated SB2 population dynamics (*not* survey results) grouped by sector (color) showing mean adult *Ae. aegypti* per house (top) and proportion of nulliparous females (bottom, Prop Null Fem).** Simulations include insecticide spraying and destructive sampling, which match field experiments as closely as possible (see Gunning et al. [32] for details). Shading shows ensemble 95% prediction interval (PI) for each group; lines within shading show ensemble median of means for model states (rows). Each ensemble (one per year) was comprised of 100 runs; for each run, daily means were computed by group. Vertical dashed lines shows spray events; lower horizontal lines show circuit durations by sector. Steep declines in adult densities early in each experiment coincide with C1 container surveys. Note that C1 was delayed in the buffer sector of S-2013. All simulations include a 1-year burn-in period (not shown). Spray efficacy was set to the empirically observed control cage mortality: S-2013, 91%; L-2014, 72%.

In SB2, we revised the spatial representation of houses to simulate key experimental events, including per-house destructive sampling of adults and immature insects within containers (i.e., eggs and larvae), as well as the impacts of indoor ULV spraying on adults. SB2 also updates the user interface to improve error-checking and streamline specification, running, and evaluation of various model configurations. Both SB and SB2 source code are released under an open source GPL 3 license, and are publicly available at https://github.com/helmingstay/SkeeterBuster/.

As in Gunning et al. [32] and SB, the house is the primary spatial unit in SB2. The relative location of houses within blocks specifies the network through which adult mosquitoes move. Each house is directly connected to zero or more neighboring houses based on geographical distance. In addition, houses are grouped by block, and each block is connected to zero or more neighboring blocks. In local dispersal, adults move among neighboring houses within a block. Between-block dispersal allows an adult to move to a random house in a neighboring block. Unlike Magori et al. [18], SB2 does not include a long-range dispersal term, though adults may traverse the simulated area via multiple steps of between-block dispersal.

We note that, while SB and SB2 were parameterized using field data from Iquitos, its performance has also been evaluated in Buenos Aires, Argentina [19, 23]. SB2 employs the standardized Global Surface Summary of the Day (GSOD) weather file format; daily weather data is

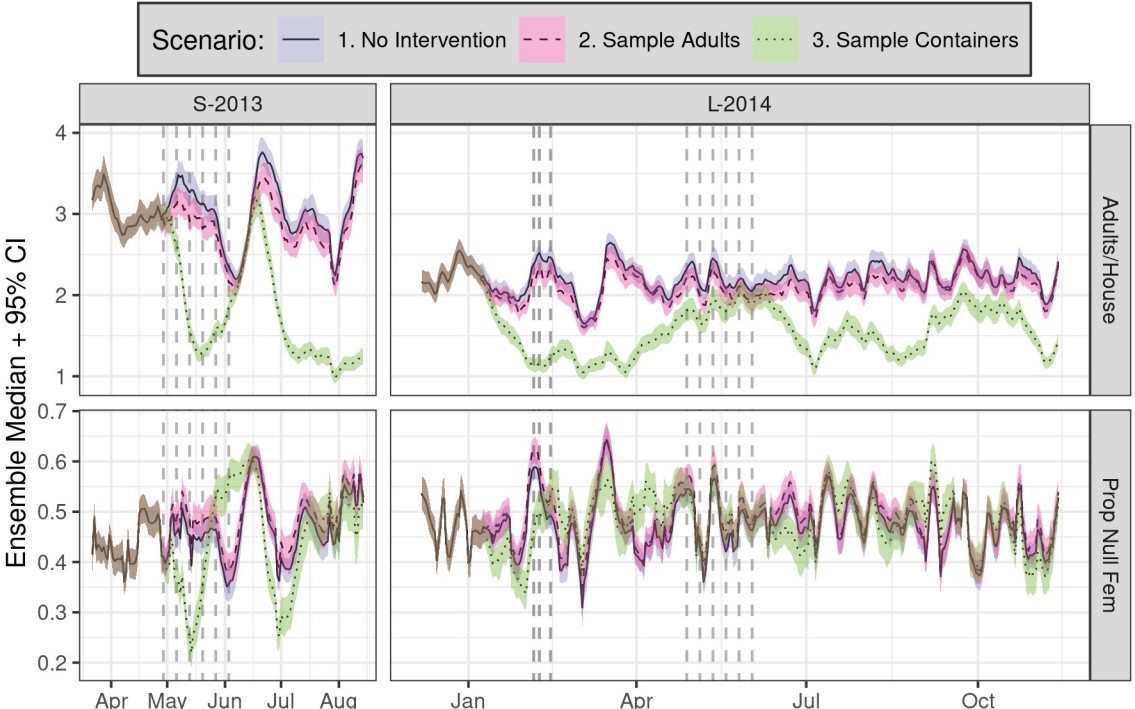

**Fig 4. Daily time series of simulated SB2 population dynamics (ensemble median of spatial means + 95% PI, 100 runs per scenario) for the (unsprayed) buffer sector of each experiment (columns) showing mean adult *Ae. aegypti* per house (top) and proportion of nulliparous females (bottom, Prop Null Fem).** Color indicates three distinct scenarios: no intervention, adult sampling only, and container sampling only. Vertical dashed lines show spray periods. Note that experimental container sampling was not conducted during spray periods, which is reflected in scenario 3.

available worldwide for most commercial airports (and many other locations, see https://cran.r-project.org/web/packages/GSODR/vignettes/GSODR.html). In addition, SB2 is distributed with a set of test scenarios that can be modified to construct complex spatial configurations and to test specific hypotheses.

## Simulation configuration

We generated spatial configurations for the simulation of each field experiment from a suite of GIS files of Iquitos (as of Jan 2016). The number and configuration of containers within each house were specified by initial baseline surveys from each experiment [32]. Consequently, cryptic larval habitat was not simulated. A summary of the resulting configurations, including total number of houses and containers, containers per house, and container-free houses by sector, is shown in Table 1 (see also S1 Table and S1 Text).

The spatial distribution of container food input is shown in Fig 2 (see also S1 Fig). To determine the food input scaling factor, we minimized the difference between field and simulation of the mean adult population across the 2013 and 2014 buffer sectors. The result is a per-container mean food input of 0.64, which yields a per-house mean food input of 1.58 and 1.15 for S-2013 and L-2014, respectively. Key model parameters are shown in S2 Table. Weather observations were taken from Coronel Francisco Secada Vignetta International Airport (Station 84377099999, see S2 Fig). For additional details, see S1 Text.

We sampled from the simulation outputs as if we were conducting empirical survey circuits of the experimental areas. This provided data at a comparable scale to those from the

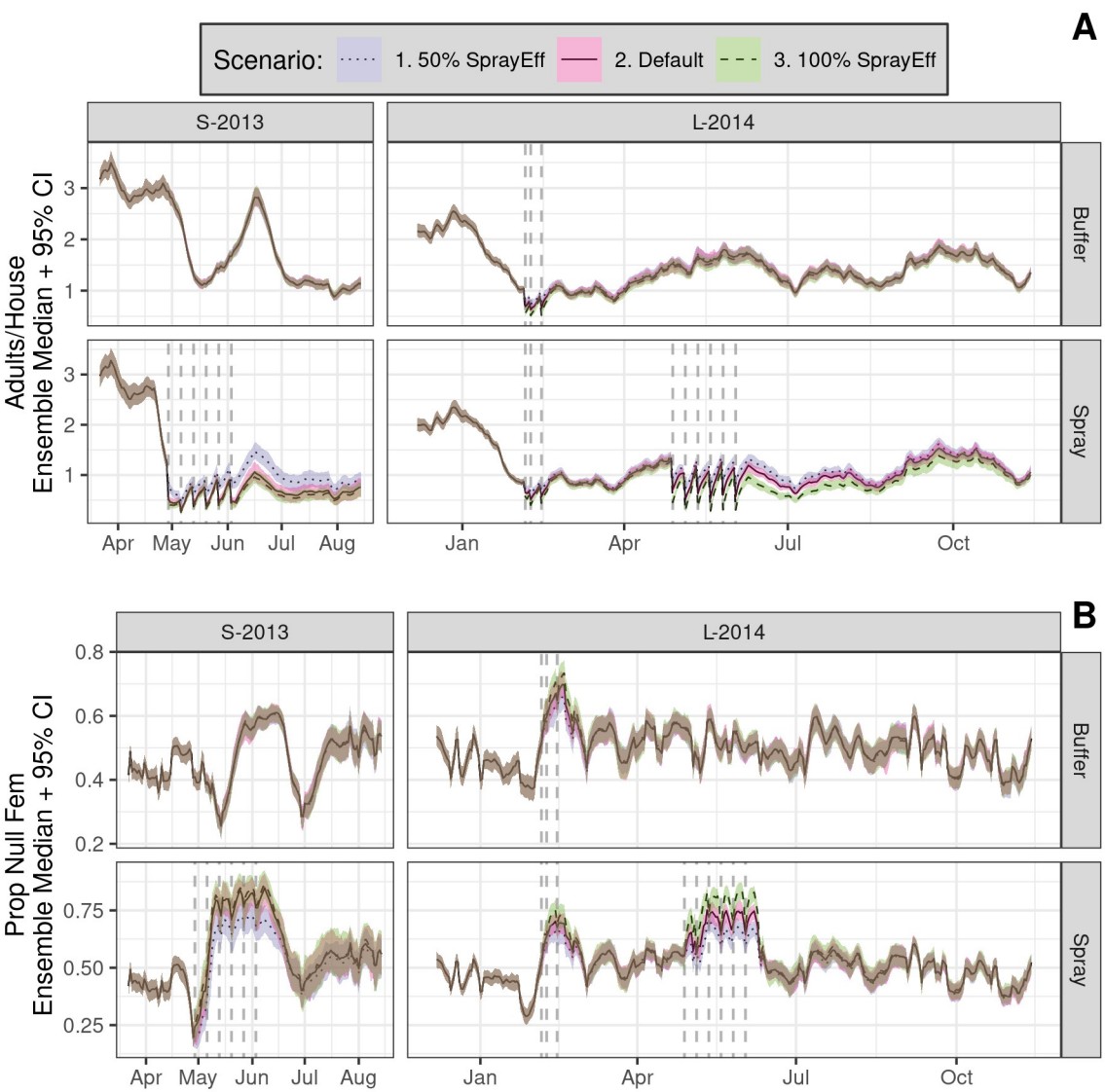

**Fig 5. Daily time series of simulated SB2 population dynamics (ensemble median of spatial means + 95% PI, 100 runs per scenario), comparing different spray efficacies (color) between sectors (rows within each panel).** Vertical dashed lines show spray periods. **A**, Adults/house. **B**, Proportion nulliparous females.

**Table 1. Summary of each simulation's spatial configuration, generated from the baseline surveys of corresponding field experiments (see S1 Text for details).**

| Experiment | Sector | Houses | Container-Free Houses | Containers | Containers/House |
|---|---|---|---|---|---|
| S-2013 | All | 1240 | 152 | 3082 | 2.485 |
| S-2013 | Buffer | 824 | 89 | 2109 | 2.559 |
| S-2013 | Spray | 416 | 63 | 973 | 2.339 |
| L-2014 | All | 2255 | 441 | 4240 | 1.880 |
| L-2014 | Buffer | 1099 | 213 | 2092 | 1.904 |
| L-2014 | Spray | 1156 | 228 | 2148 | 1.858 |

empirical surveys. The timing and location of simulated surveys and spray events was based on detailed records of the two field experiments [32]. In L-2014, the exact date of 164 spray events within the spray cycle was unknown (3.3% of 4,840 events, excluding MoH spraying); here, the onset of the spray cycle was instead used. Adult capture probability (equal to 0.29) was taken from previously estimated sampling exhaustion curves using Prokopack aspirators in household contexts [41]. We assume uniform binomial sampling of adult mosquitoes, i.e., that adults were captured independently and with equal probability regardless of time, location, sex, or age. For each experiment, spray efficacy was taken as the mean observed control cage mortality: 91% (S-2013) and 72% (L-2014) [32]. Key simulation parameters are listed in S2 Table.

## Model scenarios

We present several different model scenarios. For each scenario, an ensemble of 100 simulations was run; each run simulated and discarded one year of burn-in time. Simulations began with the onset of surveying and concluded one week after the conclusion of surveying.

We first developed a reference scenario based on detailed records from field experiments, including container surveys, adult surveys, and spray events. We used this scenario to compute a single multiplicative scaling factor of container food input that calibrated the observed mean adult population density in the buffer sector (averaged across both experiments) between simulation and experimental results. Except where noted, all other parameters were as described in previous work [18].

To better characterize model behavior, we developed six additional scenarios that selectively modified the reference scenario. To compare the expected impact of field surveys versus spraying, we added scenarios with: (1) no intervention, (2) only adult sampling, and (3) only container sampling. To assess the expected impact of weather and climate, we added a scenario that ran for 10 years (2000 to 2010, plus burn-in) with no spraying or surveying. Note that in the above four scenarios, we only consider the expected (unobserved) population dynamics. Finally, to assess the impact of spray efficacy, we added two scenarios that artificially varied spray efficacy to a low and high value (50% and 100%, respectively).

## Data analysis

Overall, we focus on variation between simulation runs within an ensemble, rather than on variation between houses within a given simulation. Variation within each ensemble is represented as 95% prediction intervals (PI).

**Expected dynamics in unperturbed conditions.** We first provide an overview of simulated *Ae. aegypti* dynamics reported by day, assuming non-destructive and complete observation of mosquitoes, including mean adults per house (Adult) and proportion nulliparous females (Prop Null). While this detailed view can provide useful insight into SB2's dynamical behavior, these results cannot be directly observed in the field due to logistical constraints, and thus are not falsifiable.

For the long-running intervention-free scenario, we compute the correlation between weather and *Ae. aegypti* populations over the full time period, stratified by mosquito life-stage. This allows us to characterize the expected long-term response of mosquito populations to temperature, humidity (i.e., saturation deficit), and precipitation.

**Comparison to field experiments.** For simulated sampling of houses (i.e., circuits), we report results grouped by experiment, sector, and circuit. Simulated sampling results can be directly compared to field experiment results [32], and are the primary focus of this study. In particular, we consider the mean number of sampled *Ae. aegypti* adults per house (AA/HSE),

Aedes house index (AHI, proportion of houses with sampled adults), the sample proportion of adult females that are nulliparous (PrNF), and positive containers per house (PC/HSE, number of sampled containers with immatures present). Note that container measures are only available in circuits where container surveys were conducted. To help characterize the immediate response of adult populations to spraying, we also provide a fine-grained weekly summary of AA/HSE during the experimental spray period for both experiments.

We compare field and simulation results in several ways. First, we compute the difference between field and simulation results, and report the ensemble distribution of differences for each measure. In order to assess the effects of intervention on each measure, we also report the ratio between the buffer and spray sectors (Spray/Buffer) within each circuit, as well as the ratio between circuits (Circuit/Baseline) within the spray sector. These ratios are directly comparable with the respective contrasts shown in Fig 5 of Gunning et al. [32], where confidence intervals for empirical results are provided. We omit empirical statistical confidence intervals here because they are not directly comparable to simulation prediction intervals.

## Results

### Overview

We begin with an overview of our simulation's internal dynamics (Figs 3–5). Next, we summarize the ensemble distribution of simulated surveys of adult and immature mosquitoes (Fig 6) that, as described in the methods, were designed to mimic the empirical data collection. We then describe the spatial distribution and time course of simulated surveys during and after spray events and compare simulated survey results with empirical survey observations (Figs 7 and 8). Finally, we examine the effects of varying spray efficacy on adult densities (Fig 9). Unless otherwise noted, simulation survey results show ensemble summaries rather than individual model runs within an ensemble.

### Simulation dynamics

The SB2 reference scenario was developed to simulate the S-2013 and L-2014 field experiments as closely as possible. Our simulations incorporate the spatial location of houses and their containers, as well as the observed location and timing of empirical field sampling and spraying (for details, see Fig 3 in Gunning et al. [32]).

We first inspect the simulated (but not empirically measured), spatially-averaged population dynamics within each sector for the reference scenario (Fig 3). Here we find that the first cycle of simulated spraying (e.g., onset of S-2013 C2 and L-2014 C6) yielded sharp and immediate declines in spray sector adult populations in both years. Yet only modest *additional* decreases were evident during the subsequent five spray cycles (Figs 3–5, vertical dashed lines). These experimental spray cycles also correspond with sharp increases in the spray sector proportion of nulliparous females that were relatively short-lived and greater in S-2013 than L-2014. As expected, the simulated effects of citywide spraying during L-2014 (C3) were approximately equivalent between sectors. This reference scenario also demonstrated considerable variation in adult populations over time, while variation among runs within the ensemble was modest by comparison. This high variability within but not between simulations suggests a large role of weather relative to demographic stochasticity (see below).

In the reference scenario (Fig 3), substantial temporal variation in adult populations was evident in the buffer sector in both years, despite a lack of spray interventions. A close inspection of buffer sector dynamics during additional scenarios that incorporate either 1) no intervention, 2) only adult sampling, or 3) only container sampling (Fig 4) indicates that container sampling yielded substantial drops in adult populations in both years. The effects of container

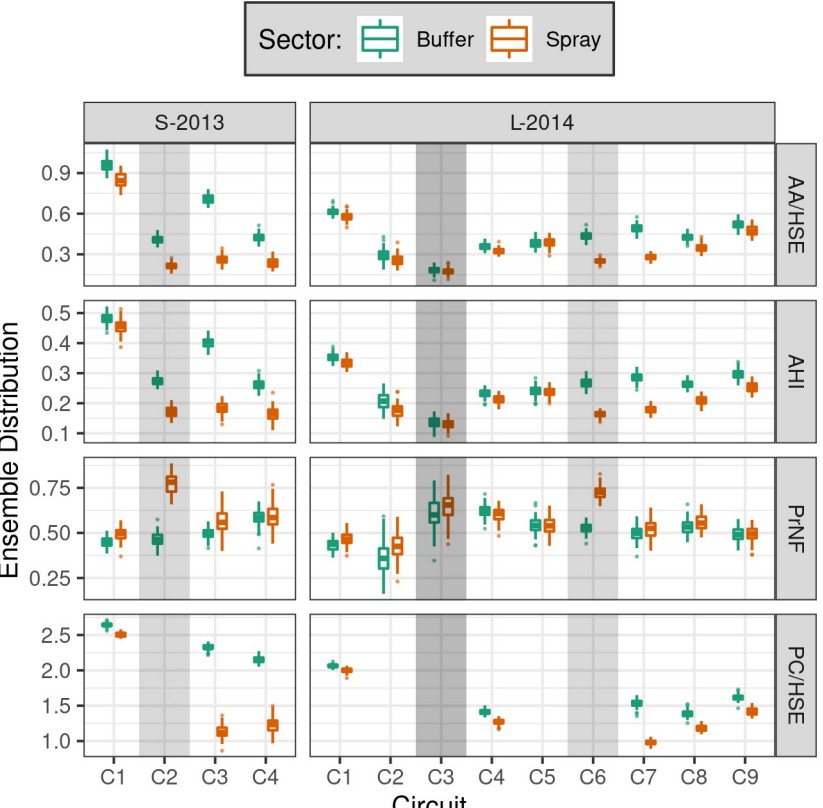

**Fig 6. SB2 simulation results of adult and container surveys.** For each measure (row), sector (color), and circuit, a per-run mean was computed across all houses; boxplots of ensemble means are shown (reference scenario, 100 runs). Vertical gray bars show periods when the spray sector was sprayed, except during L-2014 C3 (dark gray), when both sectors were sprayed. **AA/HSE**: *Ae. aegypti* adults per house (sampled). **AHI**: Adult House Index. **PrNF**: Sample proportion nulliparous females. **PC/HSE**: Positive containers per house (sampled). See S4 Fig for a comparison with empirical results.

sampling should be interpreted with caution, however, given the absence of cryptic larval habitat from field surveys and thus from our simulations (an important point we return to later).

Container sampling also perturbed the buffer sector proportion of nulliparous female adults (Fig 4), particularly in S-2013, though the effects are modest relative to temporal variation. In addition, the simulated effects of adult sampling were modest relative to within-ensemble variation. Finally, scenarios that vary spray efficacy from 50% to 100% (Fig 5) showed suprisingly modest impacts on adult densities and proportion nulliparous females relative to overall temporal variation.

To better characterize *Ae. aegypti* population dynamics in Iquitos in the absence of control measures or human intervention, and across numerous years of varying weather, we also simulated a long-running scenario (2000–2010). Here we found that the long-term dynamical response of simulated mosquito populations to weather were consistent with expectations: temperature, humidity, and precipitation exerted consistent but sometimes complex influences on all life stages (S3 Fig).

## Comparison of empirical and simulated results

Simulated survey results from the reference scenario (Fig 6) recapitulate the above model dynamics at the much coarser timescales of field surveys, e.g., weeks rather than days.

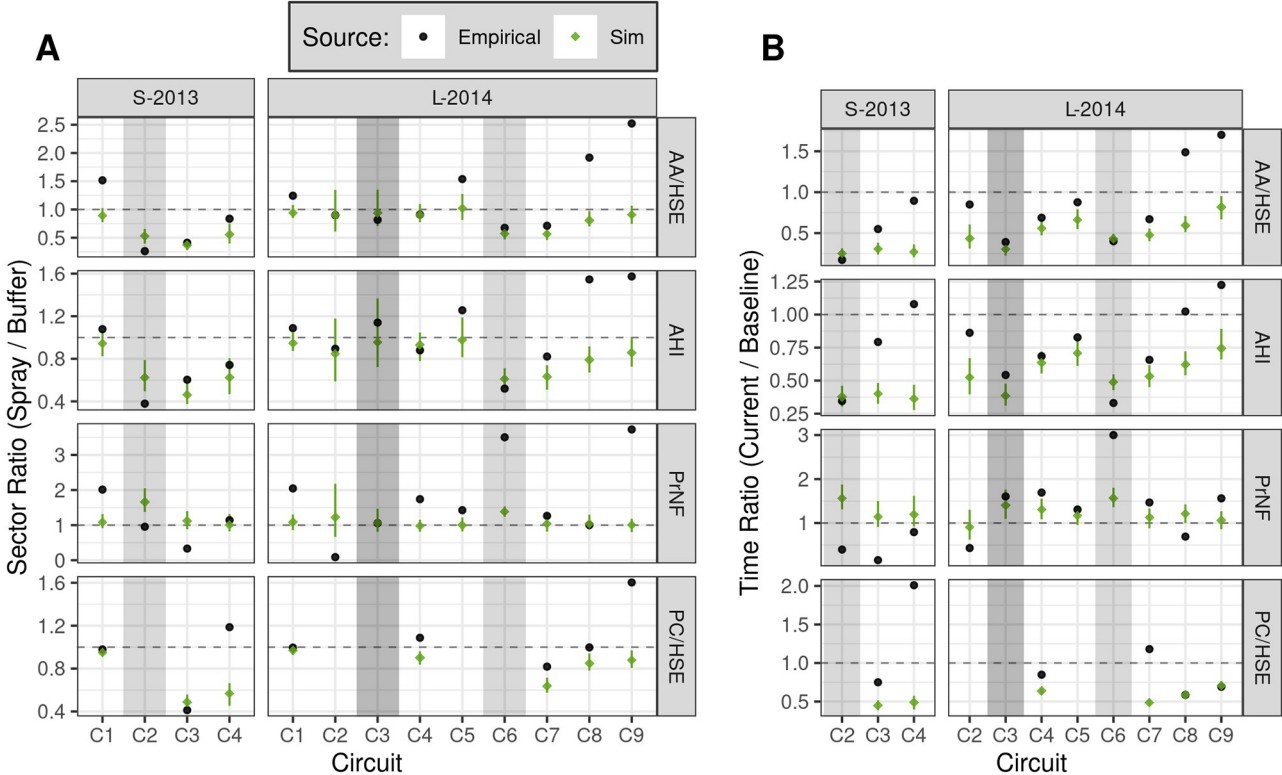

**Fig 7. Contrasts between sectors (A) and circuits (B) for each measure (row), comparing simulated and empirical surveys (select circuits).** Means were computed as in Fig 6; ratios of means are shown. **A**: ratio between sectors (spray / buffer) within each circuit. **B**: ratio of each circuit to baseline (C1) within the spray sector. Simulations show ensemble median ratio + 95% PI (reference scenario, 100 runs). Shading indicates sprayed circuits, as in Fig 6. Horizontal dashed line shows no difference (unity). See also S3 and S4 Tables. **AA/HSE**: *Ae. aegypti* adults per house (sampled). **AHI**: Adult House Index. **PrNF**: Sample proportion nulliparous females. **PC/HSE**: Positive containers per house (sampled).

Variation over time and between sectors was large relative to variation within the ensemble. The impact of container sampling on both sectors was particularly evident in S-2013, and adult abundance (as measured by AA/HSE and AHI) was reduced during and after spray events. Simulated spraying yielded a modest increase in PrNF as older adult females were killed. We observe some post-spray reduction in PC/HSE, though the sparsity of container surveys dictated by the experimental design limits our ability to detect patterns in immature insect presence.

Direct comparisons between the effects of experimental spraying in S-2013 and L-2014 are complicated by differing spray efficacy (91.2% versus 72.3%, respectively) and the time interval between spraying and subsequent adult surveys (seven days versus 1–4 days, respectively) (see also Figure 3 in Gunning et al. [32]). The relatively long intervals between field surveys makes it difficult to directly compare the empirical recovery rate, post-spraying, with the rapid recovery post-spraying seen in the full model outputs. Particularly in L-2014 (Fig 3), much of the recovery was within the interval between empirical field surveys (i.e., C7 to C8). Direct comparisons between model predictions and empirical observations are also complicated by the stochasticity and logistical heterogeneity inherent in any complex field experiment, from ULV spray applications to entomological surveys.

As in Gunning et al. [32], we describe the relative effects of spraying by contrasting between sectors (within circuit) for each measure, as well as between circuits (within the spray sector) (Fig 7). Here we compare the ratio of means between empirical results (observed point

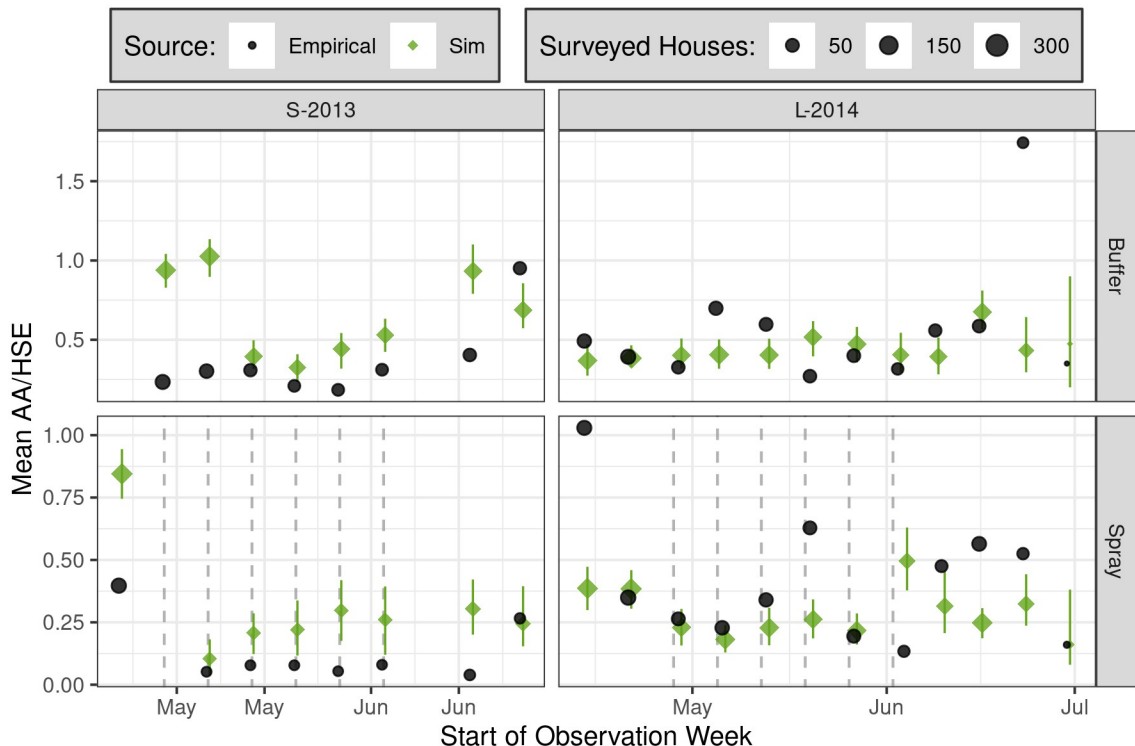

**Fig 8. Detail of weekly mean sampled *Ae. aegypti* adults per house (AA/HSE) within each sector (row) throughout experimental spraying, comparing empirical and simulated survey results (color).** SB2 simulation results show ensemble median + 95% PI. Vertical dashed lines show approximate timing of spray cycles; during each cycle, a spray attempt was made at every house.

estimate) versus simulated results (ensemble median + 95% PI). Additional details are provided in S3 and S4 Tables.

In simulations of S-2013 experimental spraying (C2, six spray cycles, 73.0–90.8% spray coverage), mean adult densities in the spray sector (AA/HSE, relative to buffer sector) were reduced by 47% (C2), 63% (C3) and 44% (C4) (Fig 7A, S3 Table). Relative to C1, mean AHI in the spray sector was reduced by 38% (C2), 54% (C3), and 38% (C4) (Fig 7B, S4 Table).

In simulations of L-2014 experimental spraying (C6, six spray cycles, 73.5–82.4% spray coverage), mean adult densities in the spray sector were reduced by 43% (C6), 43% (C7), 19% (C8), and 9% (C9) (Fig 7A, S3 Table). Relative to C1, spray sector AHI was reduced by 39% (C6), 37% (C7), 21% (C8) and 14% (C9), (Fig 7B, S4 Table). In addition, simulations of the Ministry of Health's short-duration citywide spraying during L-2014 C3 (both sectors, three spray cycles, 61.9–70.5% spray coverage) yielded modest reductions of mean AA/HSE and AHI in both sectors relative to the previous circuit (C2). We note that adults had recovered to approximately pre-intervention (C2) levels by the subsequent circuit (C4) (Figs 6 and 7B, S4 Table).

Overall, simulation results broadly agreed with empirical observations, yielding reductions in insect populations that were transient and variable in duration (Figs 7 and 8, and S4 Fig). However, simulations failed to capture initial differences in adult abundance between sectors in S-2013 (C1), and predicted a sharp drop in C2 buffer sector AA/HSE and AHI that was not empirically observed. In S-2013, simulations underestimated the immediate effects of spraying on adults in C2 (Fig 7A). Finally, simulations underestimated the speed of recovery in both years, and failed to capture the dramatic rise in spray sector adult abundances seen in L-2014

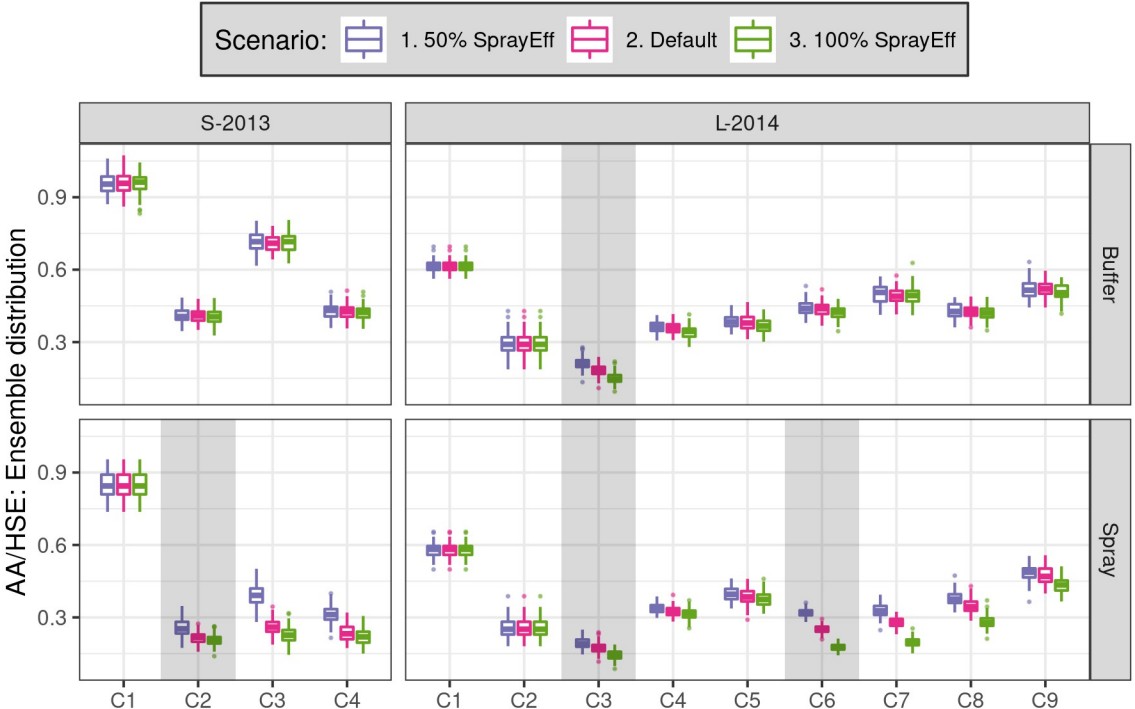

**Fig 9. Comparison of SB2 simulated *Ae. aegypti* adults sampled per house (AA/HSE).** AA/HSE within sector (row) in scenarios with different spray efficacies. Default spray efficacy was 91% (S-2013) and 72% (L-2014).

in the months following spraying (C8-C9). Substantial differences were also observed between simulated and empirical PrNF.

Due to necessary time delays between subsequent (block-stratified) field surveys, it was not generally possible to analyze insect abundance changes over short time scales (e.g., days and single weeks) across the full study area. However, weekly monitoring of adults during the experimental spray period, shown in Fig 8, reveals substantial week-to-week variation in empirical results during L-2014 that were not captured by simulations, possibly due to fine-scale operational variations in spray efficacy that were not simulated (see also Figure S3 in Gunning et al. [32]). In both years, however, simulations capture the sharp initial drop in adult densities at the onset of spraying, minimal further reductions from additional spray cycles, and a relatively rapid rebound after spraying concludes. This agreement between simulated and empirical observations is consistent with strong density dependence, a hypothesis also supported by the rapid return of adult mosquito populations to baseline levels once spraying concludes in both simulations and empirical results (Fig 7). Absent strong density dependence, we expect that repeated spraying would have continually decreased adult densities and, after spraying ceased, that adult densities would slowly recover to pre-spray levels.

## Spray efficacy

Scenarios that varied spray efficacy (Fig 9) cannot be directly compared to empirical results, yet they offer insights into the possible effects of insecticide resistance or novel insecticides. To wit, we noted a substantial decrease in empirical spray efficacy from L-2014 to S-2013 (91% to 72%) [32], and recent work has documented increases in pyrethroid resistance across two

decades in this system, culminating in a shift by the MoH in 2015 to non-pyrethroid insecticides (i.e., malathion) [31].

In S-2013, a simulated spray efficacy of 50% yielded modest reductions in spray sector adult densities in the treatment circuit (C2) relative to a much higher efficacy of 100% efficacy. In contrast, in simulations of the S-2013 spray sector, the circuit immediately after treatment (C3) yielded differences between ensembles that were small relative to between-circuit temporal variation in the S-2013 buffer sector. This suggests a modest effect of spray efficacy on the short-term control of *Ae. aegypti* populations relative to background temporal dynamics (e.g., weather) and demographic stochasticity. In L-2014, both sectors were subject to lower intensity spraying as part of the MoH's citywide intervention (C3, three spray cycles, 61.9–70.5% spray coverage). Here, modest differences are observed between 50% and 100% spray efficacy ensembles that quickly fade in subsequent circuits (e.g., C5). The effects of varying spray efficacy are larger and longer lasting during the higher intensity experimental spraying (C6, six spray cycles, 73.5–82.4% spray coverage), where differences between ensembles are still evident in C8, relative to the C3 MoH intervention.

## Discussion

The primary goal of this study was to test how well our biologically detailed model of *Ae. aegypti* population dynamics could reproduce the observed response of adult insect populations to experimental perturbations during two field experiments in Iquitos, Peru. In both simulated and empirical populations, repeated indoor ULV spraying with pyrethroids resulted in substantial yet temporary reductions in adult densities. Indeed, the proportional effects of spraying were broadly comparable between simulated and empirical results. Beyond the impacts of spraying, we found several noteworthy and unexpected differences between empirical and simulated results, particularly for PrNF and PC/HSE, as well as less temporal variation in simulated surveys of adult abundance relative to corresponding empirical observations. We also found that simulated container sampling reduced expected adult densities substantially. These results suggest that larval control could aid in vector control in the absence of cryptic larval habitat, which we do not simulate (see below). Separately, simulation results indicated a minimal impact of destructive adult sampling on mosquito populations.

### Model validation

Simulation models can assist in the design and interpretation of empirical studies and vector control efforts. However, a model's usefulness relies on the robustness and reliability of its predictions. Uncertainty quantification (e.g. [28]) can help quantify model robustness by assessing the sensitivity of model output to inevitable uncertainty and noise in initial conditions and parameterization. In addition, assessing model behavior across multiple sites, or between disparate models, can highlight important logistical constraints and reveal hidden assumptions (e.g., [19, 23]). Such model assessments, however, do not directly test models against independent empirical observations.

Model validation attempts to directly establish a model's "ground truth" by comparing its predictions to real-world observations. Such validation is, however, a particularly challenging task for detailed models that require large quantities of data to estimate their many parameters [42]. To avoid circularity, model predictions must be tested against empirical data that is independent from that used to parameterize or otherwise fit the model. In many cases, this is achieved by dividing a data set into two parts: one used to fit the model (the "training" data) and one used to validate model predictions (the "testing" set). In the case of detailed field

experiments, sufficient data for repeated out-of-sample validation would be very resource-intensive and is thus rare.

The adequacy of a model's performance may also be highly context specific, such that Oreskes et al. [43] argues for a cautious interpretation of the term "model validation". Indeed, a model may perform well when describing an unperturbed system, particularly when fit to equilibrium-state data, but perform poorly when describing the same system when perturbed. The ability of a model to predict behavior of a perturbed system is a much stronger test of its performance, and is an important consideration if a model is used to predict the outcome of a large-scale control intervention.

Motivated by these observations, we designed and implemented the two field experiments described here, where we intensively monitored real-world mosquito populations before, during, and after major perturbations. We then compared these experimental results to ensembles of simulation runs that captured demographic stochasticity. The resulting model assessment constitutes a rigorous test of our detailed simulation model, and highlights several important considerations for future vector control research.

## Comparing empirical and simulated results

Determining the key sources of disagreement between simulations and empirical observations remains an outstanding question. Our model was parameterized using the best available information in the literature and from expert opinion about *Ae. aegypti* life history and population dynamics [17, 18, 23, 28], together with more than a decade of intense field monitoring in Iquitos, Peru and elsewhere [32, 40]. In the interest of making an unbiased comparison, we have not tuned simulation parameters to match empirical results here, other than by adjusting the per-container daily food input to align simulation mean adult densities with field observations. A priori, we expect some differences between empirical and simulated results due to biotic and/or abiotic stochasticity. Disagreements could also stem from imperfect empirical observations, and from erroneous or incomplete expert assumptions about the biology and habitats of *Ae. aegypti* encoded in simulation dynamics. In addition, empirical field sampling could have been affected by a range of logistical and physical factors, including weather, survey staff, adult mosquito sex or blood-meal status. As such, attributing specific disagreements to particular simulation and/or field processes is challenging. Nonetheless, process-based simulations yield testable predictions and highlight areas of vector biology that deserve further attention, such as the role of density dependence in *Ae. aegypti* life history, and larval habitat and food availability. We explore several specific issues below.

**Female age distribution.** Adult mortality in our model is constant per unit time (i.e., Type II survivorship). Yet age-dependent survivorship of *Ae. aegypti* has been proposed based on field release-recapture field studies [44] and cage studies [45]. As others have noted [44, 46], reliable estimates of vector survivorship are difficult to obtain, but may play a key role in the epidemiology of vector-borne diseases [47, 48]. Our model provides testable predictions of adult female age distribution in response to vector control measures. Here, our simulation results differ strikingly from estimates of the absolute PrNF observed in the Iquitos field studies, while broadly reproducing the direction and duration of this response in L-2014 (Fig 7). We note that field measure of adult parity, which were based on visual inspection of dissected gonads, constitute a potential source of error, as variation in reproducibility among the field observers was evident. In addition, adult surveys, which collected mosquitoes within and around houses, could be biased with respect to female parity due to, e.g., behavior differences between parous and nulliparous females. Altogether, the observed disagreement between model output and empirical observations of PrNF deserves further attention, since it indicates

either that previous research has not fully quantified *Aedes* life history and/or behavioral ecology, or that SB2 does not properly represent the best available research.

**Density dependence and larval habitat productivity.** Competition between larvae within containers for food resources is believed to be the main process regulating *Ae. aegypti* adult population densities [49]. This competition impacts larval development time, the probability that larvae successfully complete their development, and the size of emerging adults [16, 30]. Like SB [18], SB2 explicitly tracks food resources within containers; unfortunately these dynamics (and the resulting impact on larvae) are poorly characterized in natural settings. While laboratory and cage studies have provided some insight [49, 50], exactly how the food inputs used in these studies relate to field systems remains unclear. In addition, food resource dynamics can lead to delayed density-dependent dynamics, where the current resource level of a container depends on the consumption history of current and previous larval cohorts [29, 51]. Such delayed density dependence has well-known potential for overcompensatory dynamics [52] and could explain the dramatic increase in spray sector adult densities observed late in L-2014. The conspicuous absence of a corresponding increase in the S-2013 spray sector raises difficult questions about how long an artificially perturbed system should be observed in order to rule out the possibility of a future overcompensatory response. A further complication is uncertainty surrounding the relative contribution of particular larval habitats to adult abundance [53, 54]. On one hand, our model assumes that each container's per-day larval food input scales with its surface area. On the other hand, substantial natural variability amongst containers of similar size and type is presumably found in the field. How to best measure and simulate heterogeneous container productivity is a question that deserves further attention.

**Cryptic larval habitat.** Estimating the prevalence and impact of cryptic larval habitat remains a key outstanding question in vector control, particularly in low-resource urban tropical settings. Most surveys of immature abundance in Iquitos, Peru and elsewhere focus on discrete, easily found containers [40]. However, there is increasing awareness that cryptic and otherwise overlooked larval habitats can contribute substantially to *Aedes* reproduction and vectorial capacity [54–63], and can impact the effectiveness of vector control efforts. In addition, larval habitat productivity and control efficacy can vary dramatically over both time and space, further complicating the picture [64].

Since our simulations recapitulate empirical observations, they do not explicitly incorporate cryptic (unobserved) larval habitat. This absence of *potential* habitat could explain the dramatic impact of destructive container sampling on simulated adult densities (Fig 4). We note that our empirical field trials, by necessity, *did* perturb *Ae. aegypti* populations via destructive sampling. Thus, while we cannot directly assess the impact of cryptic habitat on adult abundance here, carefully designed experimental interventions (and simulations thereof) could nonetheless yield testable hypotheses about the relative contribution of cryptic habitat to *Aedes* reproduction and vectorial capacity.

**Microclimate.** In our model, container productivity also depends on temperature and precipitation. We used observations from a single automated monitoring station (located at the nearest airport) to estimate daily average temperature and total precipitation for the entire study area, recognizing that this point measurement cannot capture the spatial variability expected of precipitation, nor diurnal variation in temperature across the diversity of houses in the two study sites. Additional sources of variation include microclimate affecting container temperature (house construction materials, vegetation, etc.). Fine-scale spatial or temporal variation in air temperature is expected to affect adult life span [65], which would affect PrNF. Detailed biophysical models have used GIS data to estimate diurnal thermal cycles in larval habitat [66], and intensive field monitoring of container water levels and temperature could yield valuable information regarding their spatial and temporal variability that could, in turn,

inform models. High-precision monitoring of abiotic drivers over long time periods across many houses, however, is not logistically feasible in most settings where *Ae. aegypti* is common.

## Conclusion

An underlying motivation for this work was a better understanding of vector-borne disease transmission of dengue virus and other Flaviviruses. Numerous models (or systems of models) have directly integrated *Ae. aegypti* population dynamics with human epidemiological dynamics at varying levels of complexity [24, 67, 68]. A more conservative approach might combine the insights of separate simulation models and empirical lab and/or field results to arrive at specific testable hypotheses regarding the influence of mosquito vector ecology and population dynamics on virus transmission dynamics. Variation in disease driven by *Ae. aegypti*'s role in virus transmission is influenced by a wide range of factors, including weather and climate [37], movement patterns of human hosts and insect vectors [69], host immune history [70], virus variation, vector lifespan [71], and vector control measures [2]. Of these many factors, our model focuses exclusively on vector population dynamics and control measures. Our targeted insights can nonetheless be used as inputs into other simulation models, or to test specific hypotheses, such as the effect of insecticidal spraying on vector age distribution and population density.

Our specific focus on Iquitos, Peru constitutes both a strength and a weakness. On one hand, wide variation in *Ae. aegypti* population biology and ecology has been observed in response to site-specific human infrastructure and cultural practices, as well as weather and climate [13, 19, 23, 56, 64, 72–74]. As such, we expect that our some of our findings are context-dependent. On the other hand, detailed study of *Ae. aegypti* in Iquitos across more than 20 years has enabled numerous iterations between surveillance, field experiments, and simulations [15, 18, 23, 31–37, 57]. Taken together, this long-running research program has yielded a nuanced picture of *Ae. aegypti* population biology and ecology in a tropical urban environment.

We have intentionally constructed SB2 to maintain compatibility with previous work that captured our general understanding of *Ae. aegypti* biology, e.g., SB [18] and CiMSIM [22]. Adding to the complexity of CiMSIM, SB incorporated spatial dynamics and demographic stochasticity, which are critical for studying invasion ecology and population genetics. SB2 was further modified to represent field studies with a high degree of spatial and temporal precision, e.g., house-based surveys and spraying. We note that SB2 (and SB) occupies an uncomfortable niche in ecological model complexity. It lacks the simplicity of mean-field models, yet makes a number of simplifying assumptions regarding insect biology and spatial processes. As such, our model is too complex for some uses (e.g., direct use by public health officials) and yet lacks sufficient detail for others (e.g., prediction). We expect that biotic and abiotic stochasticity, coupled with nonlinear population dynamics, fundamentally limit the fine-grained predictive power of any such population model. We nonetheless hope that SB2 will prove useful to identify research gaps and challenges that merit further investigation, to explore specific hypotheses about *Ae. aegypti* ecology and life history, and to design further large-scale field experiments.

The simulations presented here are based on the best available mechanistic model of two extensive and logistically complex field experiments in Iquitos, Peru. While our model's mechanistic complexity complicates attribution and interpretation, its granular structure permits a direct comparison between empirical and simulated results. This allowed us to highlight key areas of agreement, such as the effects of spraying on adult populations over time and space. We have also highlighted noteworthy disagreements where further investigation is warranted,

such as the observed proportion of nulliparous females (PrNF) or positive containers (PrPC). Our results highlight the many challenges to effective ongoing vector control, from monitoring of spray efficacy, to rapid population rebound, to the potentially long time lags between control activities and population response.

## Supporting information

**S1 Text. Technical details.**
(PDF)

**S1 Table. House counts in the baseline circuit (C1) of each experiment, showing the proportion of houses with and without successful surveys.** When available, baseline surveys were used to parameterize each house's container configuration.
(PDF)

**S2 Table. Key parameters of reference scenario.** Dispersal parameters show per-day probabilites. Container food input multiplier was determined by matching mean adult populations in the buffer sector between model and data (including both experiments together). Spray efficacy was empirically determined for each experiment from observed cage mortality: S-2013 = 0.91; L-2014 = 0.72. Finally, spray efficacy was systematically varied from the reference scenario (Low = 0.5; High = 1).
(PDF)

**S3 Table. Ratio of sector means (spray / buffer) by circuit, as in Fig 7.** Values show ensemble summary (reference scenario): **median** 95% PI (empirical). * denotes spray circuits. See also Fig 7 and S4 Table. **AA/HSE**: *Ae. aegypti* adults per house (sampled). **AHI**: Adult House Index. **PrNF**: Sample proportion nulliparous females. **PC/HSE**: Positive containers per house (sampled).
(PDF)

**S4 Table. Ratio of circuit means (indicated circuit / baseline) within the spray sector (values as in S3 Table).** See also Fig 7. **AA/HSE**: *Ae. aegypti* adults per house (sampled). **AHI**: Adult House Index. **PrNF**: Sample proportion nulliparous females. **PC/HSE**: Positive containers per house (sampled).
(PDF)

**S1 Fig. Map of simulation configuration, showing houses per polygon (top row) and containers per polygon (bottom row).** See also Fig 2.
(PDF)

**S2 Fig. Daily time series of observed weather at station 843770.** Vertical dashed lines show spray events (see Fig 3).
(PDF)

**S3 Fig. Correlation between weather and insect populations at different life stages for long-running scenario.** Adult females closely track total adult populations, while eggs and larvae have an inverse response as eggs transition into larvae. In the short-term ($<5$ days), increased temperature and humidity yield increases in adults and eggs, and a decrease in proportion nulliparous females; precipitation causes eggs to hatch into larvae, while increased temperatures cause larvae to develop into pupae. High humidity is also associated with lower larval populations across a range of time lags, most strongly at $<10$ days. In the medium to long-term (10-20 days), increased temperature and humidity was associated with increased pupal populations (peak correlation at approx. 10 days), while precipitation was associated

with increased adult populations (peak correlation at approx. 18 days). No consistent correlation between weather and number of positive containers per house is evident.
(PDF)

**S4 Fig. A**: empirical results (as in Gunning et al. [32]). **B**, ensemble residuals (simulated— empirical). See also Fig 6. **AA/HSE**: *Ae. aegypti* adults per house (sampled). **AHI**: Adult House Index. **PrNF**: Sample proportion nulliparous females. **PC/HSE**: Positive containers per house (sampled).
(PDF)

## Acknowledgments

We thank Brandon Hollingsworth, Jaye Sudweeks, Sumit Dhole, and Jennifer Baltzegar for helpful discussion. We are grateful to the Ministerio de Agricultura y Riego de Peru, Direccion General Forestal y de Fauna Silvestre for permission to conduct these studies under the auspices of Resolución Directoral Nos. 128-2007-Inrena-IFFS-DCB, 415-2009-AG-DGFFS-DGEFFS, 0022-2011-AG-DGFFS-DGEEFFS, 0330-11-AG-DGFFS -DGEFFS, and 0306-2013-MINAGRI-DGFFS/DGEFFS. We thank the residents of Iquitos, Peru for allowing us to undertake this study in and around their homes. We greatly appreciate support of the Loreto Regional Health Department including Dra. Wilma Casanova, Cristiam Carey and Hugo Rodriguez-Ferruci, and Clara Del Aguila, Raul Pinedo, Roldan Cardenas, Carlos Pacheco and Enrique Chalco of the Department of Environmental Sanitation, Peruvian Ministry of Health, Iquitos, who all facilitated our work in Iquitos. Gerson Perez Rodriguez supervised the collection and processing of mosquitoes. Entomological surveys were carried out by Jhon Bardales Cardenas, Cesar Campos Cardenas, Jimmy Maykol Castillo Pizango, Willy Chavez, Fernando Chota Ruiz, Guillermo Elespuru Hidalgo, Victor Elespuru Hidalgo, Fernando Espinoza Benavides, Rusbel Huinapi Tamani, Guillermo Inapi Huaman, Nestor Jose Nonato Lancha, Federico Reategui Viena, Edson Pilco Mermao, Angel Puertas Lozano, Juan Luiz Sifuentes Rios, Manuel Ruiz Rioja, and Abner Enrique Varzallo Lachi. Jimmy Roberto Espinoza Benavides carried out data entry. We thank Gabriela Vasquez de la Torre, Lorena Quiroz, Alfonso Vizcarra, Esther Jennifer Rios, and Jhonny Cordova Lopez, for their support in community engagement, project execution and monitoring of MoH space sprays. Drs. Robert Hontz, Christopher Mores, Frederick Stell, Craig Stoops, Diego Munoz, Cecilia Gonzales, Kyle Peterson, Adam Armstrong, Guillermo Pimentel, Zoe Moran, Toane Zuleta and Ms. Roxana Lescano of the U.S. Naval Medical Research Unit No. 6 in Lima, Peru were instrumental in facilitating these studies.

## Disclaimer

The views expressed in this article are those of the authors and do not necessarily reflect the official policy or position of the Department of the Navy, Department of Defense, nor the U.S. Government.

## Author Contributions

**Conceptualization:** Christian E. Gunning, Kenichi W. Okamoto, Fred Gould, Alun L. Lloyd.

**Data curation:** Christian E. Gunning.

**Formal analysis:** Christian E. Gunning, Fred Gould, Alun L. Lloyd.

**Funding acquisition:** Thomas W. Scott, Gissella M. Vásquez, Fred Gould, Alun L. Lloyd.

**Investigation:** Christian E. Gunning, Fred Gould, Alun L. Lloyd.

**Methodology:** Christian E. Gunning, Kenichi W. Okamoto, Fred Gould, Alun L. Lloyd.

**Project administration:** Christian E. Gunning, Fred Gould, Alun L. Lloyd.

**Resources:** Helvio Astete, Gissella M. Vásquez.

**Software:** Christian E. Gunning, Kenichi W. Okamoto.

**Supervision:** Helvio Astete, Gissella M. Vásquez.

**Validation:** Christian E. Gunning.

**Visualization:** Christian E. Gunning.

**Writing – original draft:** Christian E. Gunning, Fred Gould, Alun L. Lloyd.

**Writing – review & editing:** Christian E. Gunning, Amy C. Morrison, Kenichi W. Okamoto, Thomas W. Scott, Gissella M. Vásquez, Fred Gould, Alun L. Lloyd.

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
