## [Decision Letter · Decision Letter 0]

5 Jan 2022

Dear Mr. Gunning,

Thank you very much for submitting your manuscript "A critical assessment of the detailed Aedes aegypti simulation model Skeeter Buster 2 using field experiments of indoor insecticidal control in Iquitos, Peru." for consideration at PLOS Neglected Tropical Diseases. As with all papers reviewed by the journal, your manuscript was reviewed by members of the editorial board and by several independent reviewers. Your manuscript was positively received as being well-written and presented, and the reviewers suggest improvements to the model fitting and validation.

In light of the reviews (below this email), we would like to invite the resubmission of a revised version that takes into account the reviewers' comments. 

We cannot make any decision about publication until we have seen the revised manuscript and your response to the reviewers' comments. Your revised manuscript is also likely to be sent to reviewers for further evaluation.

Sincerely,

Lyric C Bartholomay, PhD

Associate Editor

Scott Weaver

Deputy Editor

Reviewer's Responses to Questions

**Key Review Criteria Required for Acceptance?**

**Methods**

-Are the objectives of the study clearly articulated with a clear testable hypothesis stated?

-Is the study design appropriate to address the stated objectives?

-Is the population clearly described and appropriate for the hypothesis being tested?

-Is the sample size sufficient to ensure adequate power to address the hypothesis being tested?

-Were correct statistical analysis used to support conclusions?

-Are there concerns about ethical or regulatory requirements being met?

Reviewer #1: (No Response)

Reviewer #2: This study combined modelling and field surveys, and I was impressed with the methods used for both these aspects, and also the thoughtful comparison of the two, which was the motivation for the work. See my general comment.

**Results**

-Does the analysis presented match the analysis plan?

-Are the results clearly and completely presented?

-Are the figures (Tables, Images) of sufficient quality for clarity?

Reviewer #1: (No Response)

Reviewer #2: The figures are very clear and aesthetically pleasing. The analysis was logical and sensible.

**Conclusions**

-Are the conclusions supported by the data presented?

-Are the limitations of analysis clearly described?

-Do the authors discuss how these data can be helpful to advance our understanding of the topic under study?

-Is public health relevance addressed?

Reviewer #1: (No Response)

Reviewer #2: The conclusions and limitations were well discussed. See my general comment.

**Editorial and Data Presentation Modifications?**

Reviewer #1: (No Response)

Reviewer #2: Some minor suggestions that the authors might consider are below.

 1. Can you specify the insecticide used in ULV control?

 2. How confident can you be that Prokopack aspirators collect an unbiased sample of adult female mosquitoes, with respect to their parity? Depending on their behaviour (e.g. mate-seeking, blood-seeking, resting, ovipositing etc), one might expect females are more or less readily caught by any particular capture method. If there is such a bias, this may affect the empirical estimation of nulliparous frequency – did the authors consider this? It would be helpful for this to be commented in the methods section.

 3. In the “Simulation Overview” section, a number of functional relationships embedded in the model are alluded to: “Adult movement depends on the availability of mates and larval habitat (we assume human hosts are available in all houses)... oviposition choice depends on container volume and the presence of a cover or lid, and larval development depends on weather and food availability. Survival of all life stages depends on temperature. Survival of adults and eggs also depends on water vapor pressure deficit, which is a function of temperature and relative humidity”. As a modeller but without any experience with this particular model, I wanted to know more about these assumed functional relationships. I would have particularly appreciated a figure (supplementary if the authors do not want more main manuscript figures) showing the shapes of the relationships. Having them explained in words in the section here would also be helpful.

 4. You might say something about how easy or otherwise it would be to apply the model to other settings.

**Summary and General Comments**

Reviewer #1: This paper by Gunning et al. describes a unique validation of an expanded process-based model with experimental empirical data. This is a commendable interdisciplinary combination of approaches that made a substantial and distinctive contribution to the literature and should be encouraged. The discussion of the complexity of model validation given the complex logistical constraints around entomological field data collection is also a valuable insight.

I do, however, have some comments on the fitting (calibration) and validation approaches taken that I think could be further developed to improve the manuscript.

Major comments:

Generally, the statistical procedures for model fitting and validation could be improved. Currently the model is calibrated to the data via a single multiplicative parameter (container food availability) and model fit evaluated descriptively and visually (Fig. 7-8). I understand the authors intentions about minimally modifying the model to retain generalisability, but really think that both of these should be improved. On fitting/calibration ideally multiple parameters should be selected and justified for model fitting, then a formal model fitting exercise be performed (even a simple parameter sweep within ranges suggested by previous datapoints would give more insights into the identifiability of the model when fit to data). I also think there should be some form of statistical validation of the model fit to the empirical data. This is needed to quantify the weight of evidence behind the currently suggested reasons for discrepancy between simulated and observed data and would provide a framework for the hypothesis testing that the authors claim this model will enable. 

I was surprised about the magnitude of the impact of destructive container sampling (Fig. 4) until I read about the assumption of no cryptic breeding sites. This seems like an assumption that is at least worth some sensitivity analysis as I find it difficult to believe that 100% of containers can be accurately identified and it would improve our understanding of the dynamics of this model if we could explore its sensitivity to relaxing this assumption.

Minor comments:

The figures are difficult to interpret in isolation and some changes could be made to the legend or figure text to improve interpretability, e.g. define acronyms in the legend text. Also y-axis labels for figures 1-5- what does “Adults” (range 1-3) refer to- adults per some unit of area, relative to some particular timepoint?

Reviewer #2: Though numerous models of mosquito vectors of disease have been published in recent years, few have been compared against real data. In making such a comparison, this study is useful for two reasons. First, there is an immediate benefit to validating the specific model, called SK2, giving confidence in its ability to provide insight for programmes of Aedes aegypti control. Second, the study provides a lesson in the potential, and also challenges, of validating mosquito population models with data. I was impressed by the honest discussion of the models abilities and shortcomings from the validation process. The manuscript is clearly written and the figures are instructive and visually appealing. Overall I have no hesitation in recommending publication.

PLOS authors have the option to publish the peer review history of their article (what does this mean?). If published, this will include your full peer review and any attached files.

Reviewer #1: No

Reviewer #2: No
---

## [Editor Report · Decision Letter 1]

11 Jul 2022

Dear Mr. Gunning,

Thank you very much for submitting your revised manuscript "A critical assessment of the detailed Aedes aegypti simulation model Skeeter Buster 2 using field experiments of indoor insecticidal control in Iquitos, Peru." for consideration at PLOS Neglected Tropical Diseases. The reviewers appreciated the attention to an important topic. We are very likely to accept this manuscript for publication, providing that you modify the manuscript according to the recommendations below. 

Sincerely,

Lyric C Bartholomay, PhD

Academic Editor

Scott Weaver

Section Editor

Dear Colleagues,

Thank you for your careful consideration of reviewer comments. Herein I provide some final suggested changes for your manuscript, listed according to manuscript section. I provide an excerpt from the text, followed by my comments which are preceded by >>.

Summary.

The Aedes aegypti mosquito is commonly found in tropical urban areas, and is the primary vector of several serious human pathogens, including the yellow fever, Zika, and dengue viruses.

>>consider removing ‘the’ preceding Aedes aegypti and yellow fever for readers in the PLoSNTD audience.

Methods.

Figure 1: Map showing study areas within Iquitos, Peru. Color shows sector. The larger, northern L-2014 study area borders an abandon air strip on its northwest edge. Base map by Stamen Design, under CC BY 3.0; data by OpenStreetMap, under ODbL. 

>>Please add to Figure legend to indicate years when spray event experiments were conducted at each site, and edit spelling error for ‘abandon’

Field Studies.

“We enumerated the number of adult mosquitoes per house by species. In addition, the parity of adult Ae. aegypti females was assessed post-collection by dissection and inspection of ovaries.”

>>Please provide a reference to methods for parity assessment after this sentence, in the interest of enhancing the repeatability of your work and making it accessible to researchers who have not used this method: 

Spray Intervention.

During scheduled experimental spraying, small screened cages, each containing 25 adult Ae. aegypti from a recently collected laboratory colony (Gunning 2018) 

>>please specify sex of adult mosquitoes placed in screened cages.

When mortality was ¡80%,

>>note unnecessary punctuation here.

Simulation Overview.

In brief, SB and SB2 model individual adult female Ae. aegypti, and cohorts of adult males and those of immature stages: eggs, larvae, and pupae. Adult Ae. aegypti mosquitoes (henceforth adults) dwell within and move among houses, which contain larval habitat (containers). Adult movement depends on the availability of mates and larval habitat (we assume human hosts are available in all houses). Adult females mate and subsequently lay eggs in water-filled containers, where container oviposition choice depends on container volume and the presence of a cover or lid, and larval survival development depends on food availability. Survival of all life stages depends on temperature. Survival of adults and eggs also depends on water vapor pressure deficit (VPD, a function of temperature and relative humidity), with mortality increasing at low VPD. Finally, SB2 models per-house destructive sampling of adults and immature insects within containers (i.e., eggs and larvae), as well as the impacts of indoor ULV spraying on adults

>>In this section, it would be helpful to provide references for the life history traits described herein, with particular reference to data from Ae. aegypti in Iquitos.

See sentence "We note that, while SB and SB2 were parameterized using field data from Iquitos, its performance has also been evaluated in Buenos Aires, Argentina <insert ‘by’> Legros et al. [19, 23]." 

>>missing word or parenthesis - please see <> above

>>Please check that Table 1 is as intended. The legend indicates that the table is a summary of the model configuration, but it appears to be a summary of the data used, and does not appear to speak to other elements of the caption as provided. The table is more correctly described in text, which reads “A summary of each field experiment’s configuration, including total number of houses and containers, containers per house, and container-free houses by sector, is shown in Table 1 (see also Table S1).”

 This is the current caption, which doesn't align with the table/image provided: "Table 1: Summary of model configuration. To determine the food input scaling factor, we minimized the difference between field and simulation of the mean adult population in the 2013-2014 buffer sector. The result is a per-container mean food input of 0.64, and a per-house mean food input of 1.6 and 1.16 for S-2013 and L-2014, respectively. See Table S2 for parameter details. Experiment Sector Houses Container-Free Houses Containers Containers/House S-2013 All 1240 152 3082 2.485"

Results.

Simulation Dynamics.

>>Please check references to figures to be sure that the text connects to and describes each Figure. For example, the text reads “The SB2 reference scenario was developed to simulate the S-2013 and L-2014 field experiments as closely as possible. Our simulations incorporate the spatial location of houses and their containers, as well as the observed location and timing of empirical field sampling and spraying (Figure 3).” I note that the title for Figure 3 is “Figure 3: Daily time series of model dynamics (not survey results) grouped by sector (color) for the default scenario” and that this isn't well aligned with the text.

Figure 3: Daily time series of model dynamics (not survey results) grouped by sector (color) for the default scenario

>>Please provide more information about “the default scenario” to make figure legends as clear and stand-alone as possible, and consider specifying that the figure depicts predicted population dynamics according to SB2.

Figure 4: Daily time series of model dynamics for buffer sector, showing three scenarios (color): no intervention, adult sampling only, and container sampling only (ensemble median of means + 95% CI). In scenario 3, container sampling was not conducted during spray periods. See Figure 3 for additional details. 

>>Please provide more information about “the buffer sector” to make figure legends as clear and stand-alone as possible. In this case, you could consider referring to Figure 1 for clarity. Also please explain what additional details can be found in Figure 3.

Figure 5: Daily time series of model dynamics (ensemble median of means + 95% CI), comparing different spray efficacies (color) between sectors (rows within each panel). A, Adults/house. B, Proportion nulliparous females. See Figure 3 for additional details.

>>consider adding additional details (e.g., Skeeter Buster 2) so figure legend stands alone; Also please explain what additional details can be found in Figure 3.

Figure 6: Simulated survey results. F

>>consider adding detail to ‘survey results’, e.g., life stage sampled or expected

Figure 8: Detail of weekly mean AA/HSE within each sector (row) throughout experimental spraying, comparing empirical and simulated survey results (color). Simulation results show ensemble median + 95% PI. Vertical dashed lines shows approximate timing of spray cycles; during each cycle, a spray attempt was made at every house. 

>>please spell out AA/HSE in figure legend. Note spelling error ‘shows’

Figure 9: Comparison of simulated Ae. aegypti adults sampled per house (AA/HSE). AA/HSE within sector (row) among different spray efficacies. See Figure 3 and Figure 6 for additional details.

>>consider adding detail to explain S-2013 and L-2014 in figure legend.

Discussion.

Our model explicitly tracks food resources within containers, but unfortunately these dynamics (and the resulting impact on larvae) are poorly characterized in natural settings.

>>This point is not clear from the Methods, which state “and larval development depends on weather and food availability.” How are food resources tracked within containers in the model? Please clarify either here or in Methods section.

Figure Files:

Data Requirements:

Reproducibility:

References

---

## [Editor Report · Decision Letter 2]

30 Aug 2022

Dear Mr. Gunning,

Thank you very much for submitting your manuscript "A critical assessment of the detailed Aedes aegypti simulation model Skeeter Buster 2 using field experiments of indoor insecticidal control in Iquitos, Peru." for consideration at PLOS Neglected Tropical Diseases. Thank you for providing careful revisions to this paper. Your manuscript reads very well and I'm looking forward to seeing it published! I have very minor suggestions for final editing, and will be happy to accept immediately upon return.

1) please check with authors to be sure names are as the authors intend. I noted that Dr. Scott, for example, usually is listed as "Thomas" in other publications.

2) The legend for Figure 3 has changed and needs slight revision to be understood. Currently it reads "Daily time series of SB2 population dynamics (not survey results) grouped by sector (color)r the , which was designed to match field experiments as closely as possible." Please edit "(color)r the , " in particular.

3) The legends for Figures 3 and 4 should be edited to include description of the top panel (adult abundance) and lower panel (parity), and a definition should be provided for "Prop Null Fem" for readers to interpret data shown.

[1] A brief description of the changes you have made in the manuscript according to the suggestions below.

Sincerely,

Lyric C Bartholomay, PhD

Academic Editor

Scott Weaver

Section Editor

Thank you for providing careful revisions. Your manuscript reads very well and I'm looking forward to seeing it published! I have very minor suggestions for final editing, and will be happy to accept immediately upon return.

1) please check with authors to be sure names are as the authors intend. I noted that Dr. Scott, for example, usually is listed as "Thomas" in other publications.

2) The legend for Figure 3 has changed and needs slight revision to be understood. Currently it reads "Daily time series of SB2 population dynamics (not survey results) grouped by sector (color)r the , which was designed to match field experiments as closely as possible." Please edit "(color)r the , " in particular.

3) The legends for Figures 3 and 4 should be edited to include description of the top panel (adult abundance) and lower panel (parity), and a definition should be provided for "Prop Null Fem" for readers to interpret data shown.

Figure Files:

Data Requirements:

Reproducibility:

References

---

## [Editor Report · Decision Letter 3]

3 Oct 2022

Dear Mr. Gunning,

We are pleased to inform you that your manuscript 'A critical assessment of the detailed Aedes aegypti simulation model Skeeter Buster 2 using field experiments of indoor insecticidal control in Iquitos, Peru.' has been provisionally accepted for publication in PLOS Neglected Tropical Diseases.

Best regards,

Lyric C Bartholomay, PhD

Academic Editor

Scott Weaver

Section Editor

---

## [Editor Report · Acceptance letter]

30 Nov 2022

Dear Mr. Gunning,

We are delighted to inform you that your manuscript, "A critical assessment of the detailed *Aedes aegypti* simulation model Skeeter Buster 2 using field experiments of indoor insecticidal control in Iquitos, Peru.," has been formally accepted for publication in PLOS Neglected Tropical Diseases.

Best regards,

Shaden Kamhawi

co-Editor-in-Chief

Paul Brindley

co-Editor-in-Chief
